# OpenReview forum: "AdaSCALE: Adaptive Scaling for OOD Detection"
_ICML.cc/2026/Conference — ICML 2026 regular_

### Official Review · Reviewer_ouro · 2026-02-20

**Soundness:** 3
**Presentation:** 3
**Significance:** 3
**Originality:** 3
**Overall Recommendation:** 4
**Confidence:** 3

**Summary:**

The paper tackles out-of-distribution (OOD) detection for deep neural networks. The paper first analyzes the previous work of scaling that adopts static threshold. Then the paper presents a new method to adaptive scaling where the percentile threshold used in scaling depends on each sample’s estimated OODness. The key observation is OOD samples show greater instability in high-magnitude activations under small perturbations compared to in-distribution inputs. Using the perturbed inputs, it formulates a adapted percentile threshold.

**Compliance With Llm Reviewing Policy:**

Affirmed.

**Key Questions For Authors:**

I may have missed the answers for the following questions, would you like to point me out or answer the questions:

Please explain briefly, not based on empirical observations but some theoretical perspectives, why the designs of the methods work better than previous methods. This can help the community to deepen the understanding of the techniques.

The paper is not self-contained as terms are not defined or cited in the paper. For example, give a clear citation of MSP, EBO etc. Define what the near-OOD tasks and far-OOD mean, at least in appendix.

With the above citation, a brief discussion of why Swin-B perform better than the BFAct will be appreciated. Does that mean some techniques in AdaSCALE could not work as well in Swin-B?

**Limitations:**

No. It might be helpful when the method might fail (for architectures etc).

**Strengths And Weaknesses:**

**Strength**

The paper is well motivated and contains a lot of empirical evidence. The design choices of the paper are empirically supported with experiments. The experiments are comprehensive and the method. The paper is clear to follow.

**Weakness**

This is an empirical paper grounded in earlier findings. The paper contains techniques to improve the performance without theoretical reasoning. For practical utility that’s enough, but not for deeper understanding of the method.

---

> ### Author Rebuttal · Authors · 2026-03-29
>
> ## Perspective on why the design works
>
> We thank the reviewer for this thoughtful question. We provide the theoretical grounding below organized around two key ideas.
>
> - **Our core OOD signal (activation shift) is a principled extension of ReAct's well-established observation.** ReAct (Sun et al., 2021) showed that OOD samples induce abnormally high activations. We extend this with the observation that the *positions* of these high activations are relatively unstable under minor perturbations for OOD samples, but stable for ID samples:
>     - *ID stability:* A model trained on ID data has learned the data manifold. Small perturbations do not drastically alter semantic features, so the highest-magnitude activations remain at stable neuron positions.
>     - *OOD instability:* The model has no knowledge of OOD data. A minor perturbation of an OOD input produces what the model perceives as a completely different unfamiliar sample. Per ReAct, both the original and perturbed OOD inputs trigger abnormally high activations.
>     - *The Shift in High Activations:* But, since they are effectively two distinct unknown inputs, the probability that these high activations land at the exact same neurons is low. This causes pronounced shifts that our method measures through Q.
>     - *Observation:* Different ID classes all share the property of being well-modeled. If Q were merely capturing inter-class variation, the ratio Q_OOD / Q_ID would fluctuate around 1. Instead, Figure 4 shows it is consistently above 1 across multiple OOD datasets, confirming systematic separation.
>
> - **AdaSCALE works because it fuses three complementary, partially independent OOD signals through an adaptive mechanism.** Prior scaling methods (Djurisic et al., 2023; Xu et al., 2024; Djurisic et al., 2024) already succeed by fusing two signals: activation patterns and original logit information. However, they impose a static percentile threshold identical across all samples, which limits effectiveness. AdaSCALE injects a third independent signal (activation shift via Q) through the percentile, making it dynamic and sample-specific.
>     - *Q is not claimed to be an oracle:* Q alone achieves only 79.81 / 72.32 (FPR@95 / AUROC) on near-OOD with ResNet-50 on ImageNet-1k. Its value is as a complementary signal. The jump from 79.81 to 59.43 FPR@95 when Q is integrated into the scaling mechanism (vs. standalone) confirms that the gain comes from complementary fusion.
>     - *This also explains why AdaSCALE outperforms OptFS:* OptFS assumes activation reshaping benefits every sample. For near-OOD inputs with ID-like activation patterns, this can distort informative logit signals. AdaSCALE avoids this by letting alternative signals compensate when one becomes unreliable for a given sample.
>
> ## Citations and near-OOD / far-OOD definition
> Thanks. We'll incorporate it.
>
> ## Discussion on Swin-B
>
> We thank the reviewer for the insightful question. We would like to clarify that AdaSCALE does in fact work well on the Swin-B architecture. The apparent advantage of BFAct is limited in scope.
>
> **AdaSCALE is competitive with BFAct in near-OOD on Swin-B.** Comparing FPR@95 / AUROC, AdaSCALE-L achieves 72.89 / 74.58 versus BFAct's 71.81 / 75.28 in the near-OOD setting, which is essentially on par. BFAct's advantage over other methods manifests primarily in the far-OOD setting and mainly on the FPR@95 metric. This does not indicate a fundamental incompatibility between AdaSCALE and the Swin-B architecture.
>
> **The adaptive component of AdaSCALE substantially improves upon the SCALE baseline on Swin-B.** SCALE (our predecessor baseline) achieves 88.62 / 61.47 (near-OOD) and 86.59 / 66.77 (far-OOD) in FPR@95 / AUROC. AdaSCALE improves over SCALE by 15.73 and 40.35 points in FPR@95 for near-OOD and far-OOD respectively, confirming that the adaptivity introduced by our method is highly effective on this architecture.
>
> **On BFAct's far-OOD FPR@95 performance on Swin-B.** We hypothesize that this may be related to the nature of BFAct's Butterworth filter-inspired shaping function, $\theta_{\text{BFAct}}(z) = 1 / (1 + (z/t)^{2N})$, which provides a smooth, gradual roll-off for high-magnitude features rather than a hard threshold (as in ReAct) or discrete interval-based rescaling. Swin-B's hierarchical shifted-window attention may produce penultimate-layer activations with a smoother magnitude distribution and the continuous attenuation of the Butterworth filter may be better suited to it. That said, we have not conducted a thorough analysis of this specific interaction, and further investigation would be needed to confirm this intuition.
>
> In summary, AdaSCALE remains competitive in near-OOD and delivers large gains over SCALE across both settings. The strong far-OOD FPR@95 result of BFAct on Swin-B likely reflects a favorable interaction between its shaping function and Swin-B's activation characteristics, rather than a shortcoming of AdaSCALE.
>
> ---
>
> We are happy to discuss further.

---

> > ### Author Rebuttal · Reviewer_ouro · 2026-04-02
> >
> > The authors have resolved my concerns. Thank you for the explanations.

---

> > > ### Author Response · Authors · 2026-04-02
> > >
> > > We thank Reviewer ouro for the thoughtful and constructive discussion. Since your concerns have been fully resolved, we kindly request if you might consider raising your score to reflect this.

---

### Official Review · Reviewer_2Ar8 · 2026-02-24

**Soundness:** 3
**Presentation:** 3
**Significance:** 3
**Originality:** 3
**Overall Recommendation:** 4
**Confidence:** 4

**Summary:**

The paper presents an OOD detection technique that adaptively scales activation of each input using the OOD likelihood obtained by subtracting the activation of original image and its perturbed variant. The proposed method is a post-hoc detection method thus eliminating the need for comprehensive training, as well as reliance on OOD regularization through outlier expsure. The experiments results are conducted across multiple OOD detection benchmarks which shows superior performance compared to existing baselines.

**Compliance With Llm Reviewing Policy:**

Affirmed.

**Key Questions For Authors:**

1. It seems that the adaptive scaling of activation map provide a good measure for OOD likelihood. Does the scaling work in logit level as well?
2. Gradient based attribution function AT is used to create a perturbed variant of x. How do you obtain the AT function? Based on Section 4.2.1, AT is derived from the gradient of f(x), so it requires an input to undergo a full forward pass. Does this mean the AT function is trained on the ID classes?
3. How does Q and C_o exhibit opposing behaviour? The formulation for them is the same except for the number of considered activations (k1, and k2). The relationship between figure 2, and 5 is not clear.
4. How do you ensure the activation shift between a and a^epsilon is necessarily the estimation of OODness? How do you quantify such a shift does not lie within the ID distribution (different classes within the ID manifold), but a indicator of OOD likelihood?

**Limitations:**

Yes

**Strengths And Weaknesses:**

Strengths:
The paper addreses a long outstanding OOD problem in machine learning using a novel approach of adaptive scaling, making their contribution highly significant and influential in this domain. The paper is written well overall and narrative is easy to follow and it includes multiple experiments, abaltaion, sensitivity analysis making it technically sound. Although the paper adopts idea from famous activation scaling technique, the method of adaptive scaling based on the ood likelihood is quite novel.

---

> ### Author Rebuttal · Authors · 2026-03-29
>
> We thank Reviewer 2Ar8 for recognizing our contribution as highly significant and influential as well as "quite novel" in the rich literature of OOD detection. We answer the queries below:
>
> ## Question 1
>
> AdaSCALE-**A** refers to scaling *activation* and AdaSCALE-**L** refers to scaling *logits*. As seen from the results in the paper and appendix, the performance is almost the same.
>
> ---
> ## Question 2
>
> The attribution function AT is **not a separately trained module**. It is simply the gradient of the pre-trained classifier's output with respect to input pixels, via standard backpropagation. AT does not need any training beyond what the base classifier has already seen. This is the same assumption shared by all post-hoc OOD detection methods (MSP, Energy, ReAct, SCALE, etc.). As shown in the paper (Remark in 4.2.1) and Appendix (perturbation study), **even random pixel selection performs comparably** to gradient-based attribution. The random perturbation variant removes the backward pass entirely, bringing latency down from $2.91\times$ to $1.56\times$ relative to SCALE (see the Latency table), while maintaining strong OOD detection results. This confirms that AdaSCALE's improvements come from the *activation shift phenomenon itself*, rather than any specific attribution function.
>
>  ---
>
> ## Question 3
>
> Q and C_o have opposing behaviour: Q is higher for OOD while lower for ID. Conversely, C_o is lower for OOD while higher for ID. This is the opposing behaviour which we deal with.
>
> ---
>
> ## Question 4
>
> We discuss the intuitive grounding, empirical verification, and the role this signal plays below.
>
> **1. Intuitive Grounding: Extending ReAct's Observation**
>
> The core phenomenon follows from well-established observations. ReAct (Sun et al., 2021) showed that OOD samples often induce *abnormally high activations*. We extend this with our observation that the *positions* of such high activations in OOD samples are relatively unstable under minor perturbations compared to ID samples.
>
> A model trained on the ID distribution has learned the data manifold, so it is robust to minor perturbations of ID inputs. Small changes should not drastically alter semantic features, leading to stable activation patterns at the highest-magnitude positions. Conversely, the model has no knowledge of OOD data. A minor perturbation of an OOD input can produce what the model perceives as a *completely different* unfamiliar input. Since both the original and perturbed OOD input are unknown, both will trigger "abnormally high activations" (per ReAct), but the probability that these land at the *exact same* neurons is low. This results in pronounced shifts at the highest-magnitude activations, which is what our method measures.
>
> Regarding within-ID variation: different classes *within* the ID manifold all share the property of being well-modeled by the network. Minor perturbations of an ID sample, regardless of its class, do not cause the same instability in peak activations that we observe for OOD inputs.
>
> **2. Empirical Verification via the Q_OOD / Q_ID Ratio**
>
> As shown in Figure 4, the ratio Q_OOD / Q_ID is consistently **greater than 1** across multiple OOD datasets on ImageNet-1k with ResNet-50, confirming that Q systematically assigns higher values to OOD samples. If the activation shift were merely capturing within-ID variation (e.g., inter-class differences), this consistent separation would not hold; the ratio would fluctuate around 1.
>
> **3. Role Within AdaSCALE: An Independent, Complementary Signal (Not an Oracle)**
>
> We do *not* claim Q (or Q') is a perfect OOD oracle. As shown in the ablation "How effective is Q without scaling?", Q alone achieves only 79.81 / 72.32 (FPR@95 / AUROC) on near-OOD detection with ResNet-50 on ImageNet-1k, far from sufficient on its own. Its value lies in providing an *independent and complementary* OOD signal distinct from the two signals already fused by existing scaling methods: (i) activation patterns and (ii) original logit information. By injecting this third source of evidence *through* the adaptive percentile mechanism, AdaSCALE allows the three signals to compensate for each other's weaknesses. The improvement from 79.81 to 59.43 in FPR@95 when integrating Q into the scaling mechanism (rather than standalone) confirms that its contribution is realized through complementary fusion, not standalone discriminative power.
>
> To summarize: the activation shift is a principled and empirically validated OOD indicator, verified by Q_OOD / Q_ID being consistently above 1, and effective not as an oracle but as an independent signal strengthening the existing scaling framework.
>
> We hope these clarifications resolve the reviewer's concerns and are happy to discuss further.

---

> > ### Author Rebuttal · Reviewer_2Ar8 · 2026-04-01
> >
> > Thank you authors for the response to my comments. Regarding the formulation of C_o: Why is the perturbed activation choosen instead of the original activation while doing ReLU? Based on the intuition that Q, and C_o exhibit opposing behaviour and the latter is higher for ID, shouldn't C_o capture change in the original input (activation or logit) rather than on its perturbed variant (activation or logit)?
> > I think that higher C_o for ID should only be possible if the descrepancy is captured (by ReLU) for ID.

---

> > > ### Author Response · Authors · 2026-04-02
> > >
> > > Thank you for the follow-up. We believe there is a misunderstanding about the formulations of $Q$ and $C_o$. They are **not** the same operation with different $k$ values. They compute fundamentally different quantities:
> > >
> > > - $Q = \sum_{j \in \text{argsort}(\mathbf{a}, \text{desc=True})[:k_1]} (|\mathbf{a}^\varepsilon_j - \mathbf{a}_j|)$ sums the **activation shift** (absolute difference) at the top-$k_1$ positions. This is higher for OOD because OOD activations are unstable under perturbation (Figure 2).
> > >
> > > - $C_o = \sum_{j \in \text{argsort}(\mathbf{a}, \text{desc=True})[:k_2]} \text{ReLU}(\mathbf{a}^\varepsilon_j)$ sums the **perturbed activation magnitudes** at the top-$k_2$ positions. Note that this **is not** a difference/shift at all. This is higher for ID because ID samples maintain high activation magnitudes even after perturbation (Figure 5).
> > >
> > > The opposing behavior comes from a completely different empirical phenomenon: activation magnitude preservation (Figure 5) vs. activation instability (Figure 2). $Q$ asks "how much did the activations shift?" while $C_o$ asks "how large are the perturbed activations?". These are two different questions, and their answers happen to go in opposite directions for ID vs OOD samples.
> > >
> > > Regarding why we use perturbed activations $\mathbf{a}^\varepsilon$ rather than original activations $\mathbf{a}$ in $C_o$: Table 13 directly addresses this. Row 6 shows the variant using original activation magnitudes, while row 4 shows our formulation using perturbed activations. Both work well, and the difference in performance is quite small (e.g., 17.84 vs 18.02 FPR@95 in far-OOD). In practice, either choice is reasonable; we went with the perturbed variant simply because it performed slightly better in our ablations (Table 13), but this is a fairly trivial design choice since both options lead to comparable results.
> > >
> > > We thank Reviewer 2Ar8 for their thoughtful engagement and hope the above clarification resolves the concern. If it does, we request for the consideration of increasing the score.

---

### Official Review · Reviewer_JuwQ · 2026-03-06

**Soundness:** 2
**Presentation:** 1
**Significance:** 2
**Originality:** 2
**Overall Recommendation:** 2
**Confidence:** 5

**Summary:**

The paper introduces an adaptive scaling method for activations that uses an input-dependent percentile threshold based on the likelihood of OOD samples. The authors observe that OOD samples exhibit a higher activation shift (particularly in the top activations) compared to ID samples under minor perturbations. Quantifying this activation shift results in a higher threshold for ID samples compared to OOD samples. Through extensive evaluations, the authors demonstrate superior performance.

**Compliance With Llm Reviewing Policy:**

Affirmed.

**Final Justification:**

There is a limited budget of 7000 characters, so I will keep only high value concerns below.

The best part of this paper is the extensive evaluation. The sheer volume of evaluation is worthy of high appreciation. The rebuttal response successfully clarifies minor clerical issues. However, I believe that in reputed conferences like ICML, novelty, proper positioning, and transparency add more value to the contribution.

$\textbf{Incorrect Presentation/Intuition}$ In Section 4.1 (Pros analysis, Insights), activation patterns and original logits are not independent. They may be complementary, but not independent. Even in the rebuttal response to Reviewer 2Ar8, the authors claim them to be independent without any basis, proof, or evidence. The well-established observation from ReAct is largely limited to the ResNet family, which uses Batch Normalization (BN), and may not extend to more recent architectures. This assumption does not hold for ConvNeXt, ViT, or Swin, where feature vectors do not exhibit abnormally high activations for OOD samples. While the mechanism may still work, the underlying intuition used for motivation appears to be incorrect. Additionally, what is $\bar{Q'}$ in Eq. 6?

$\textbf{Accuracy Claim}$ AdaSCALE-A alters/scales the activation values that pass through the classification head to obtain logits. Mathematically, it is possible that the classification order might change. Although it can be proven that the classification ordering remains intact in AdaSCALE-L, the paper does not provide this justification, and the rebuttal does not clarify it concretely. Moreover, the accuracy claim should be supported by explicit accuracy evaluation.

$\textbf{Incremental Novelty}$ The majority of the design components presented in the paper already exist in the literature. These include perturbation (selective pixel perturbation as opposed to ODIN’s entire image), gradient attribution to select input pixels (similar to the selection set R in BAT and LAPS, but at the pixel level instead of the activation level), and exact same scaling frameworks as SCALE and related methods. The authors claim that “gradient attribution is not even needed,” which raises the question of why such complex and computationally expensive operations are introduced, as the paper would arguably be better without them. Moreover, the paper does not clearly explain what is adaptive about AdaSCALE (the adaptive component appears to be the use of eCDF). The novelty primarily lies in the integration and engineering of these high-level ideas, which does not justify the strong novelty claims made in the paper. The rebuttal responses regarding novelty largely reflect concepts already present in the literature. I believe ICML has a higher threshold for novelty.

$\textbf{SOTA Claim}$ The blanket claim of SOTA performance is unfair and misleading. On overlapping CIFAR evaluations, CORES outperforms AdaSCALE using both DenseNet and WRN (e.g., on CIFAR-10 with WRN, CORES achieves 6.03 FPR compared to 64.96 for AdaSCALE on Texture). On ImageNet evaluations, CADRef demonstrates better generalization than OptFS. The relatively poor performance of AdaSCALE on CIFAR raises concerns that the strong performance on ImageNet may be due to extensive hyperparameter tuning. Moreover, on vision transformer architectures, BFAct performs better (as shown in their own experiments). The authors justify comparisons against SCALE based on its strong performance on ResNet-50; however, they themselves acknowledge that SCALE performs poorly outside the ResNet family (while good performance on EfficientNet is a strength of the method, it should not be mentioned limitations). The issue is not that SCALE or ASH perform poorly on modern architectures, but that comparing against them despite their known limitations to claim SOTA performance is concerning.
Overall, the blanket SOTA claim made in the paper is unfair and misleading, especially when strong methods such as CORES and CADRef are not included in the comparison. More careful and accurate positioning is needed.

$\textbf{Analysis}$ Major methods such as Energy, ODIN, GradNorm, GradOrth, ReAct, DICE, and SCALE (which also explains why ASH works) provide detailed analysis. The rebuttal claim that “AdaSCALE operates in the same empirical tradition as its predecessors, none of which provide formal theoretical guarantees, yet all are widely accepted contributions in this space” misrepresents these works and is highly inappropriate. In my view, given that adaptive scaling builds heavily on prior approaches for OOD detection, the paper should provide theoretical/statistical insights into the proposed method, rather than relying primarily on intuitive reasoning and empirical results. This would help justify the design choices more rigorously and strengthen the contribution. The rebuttal did not address this concern, apart from mentioning it as future work.

$\textbf{Why is analysis important?}$ The authors motivate and introduce the use of correction terms in Eq. 5. They also use ResNet-50 to fix all the non-percentile parameters. However, in Table 12, the performance gain on ResNet-50 is not significant compared to other models. This raises the question of how the authors concluded that the correction term would produce strong results on other architectures (except ResNet-50). Unless the authors evaluated all models before hand and observed that the correction term is important, this introduces a concern. If such evaluations were used to guide the design, it may indicate data peeking or potential leakage. An analysis on why correction terms are important is necessary; it could eliminate the need for extensive OOD detection evaluation

$\textbf{Hyper-parameters}$ The author’s denial of counting non-percentile hyper-parameters as tunable is concerning. These parameters are tuned using ResNet-50 and OpenImage-O (val split), which implies the use of an OOD dataset. Therefore, the characterization of tuning “only one hyper-parameter” is misleading, especially when the results are reported with extensive tuning of $(p_{min}, p_{max})$. The author’s claim that this is “consistent with the literature” is incorrect and misleading. Most major techniques do not involve this level of tuning; in fact, they typically fix hyper-parameters, whereas the authors do not acknowledge their non-percentile parameters as tunable hyper-parameters. Moreover, many established methods (with some recent exceptions) explicitly use auxiliary OOD datasets for hyper-parameter tuning. Otherwise, given access to OOD data like OpenImage-O (val split), more sophisticated approaches from semi-supervised learning could be applied. Finally, if fixing $p_{min}$​ can already achieve near-optimal performance, it raises the question of why such extensive tuning is necessary. A simpler setup with fixed hyper-parameters would likely provide better generalization.

Based on these critical flaws, which still persist in the paper, and the authors’ refusal to address them, I strongly recommend rejecting this paper and encouraging resubmission after major revision. The paper would benefit from proper positioning of its contribution, clearer disclosure of hyper-parameter selection, and more restrained SOTA claims. With these changes, the contribution would better serve the research community.

I focus my evaluation on the technical aspects of the paper and defer any non-technical concerns to the AC and PC.

**Key Questions For Authors:**

See Weaknesses.

**Limitations:**

Yes

**Strengths And Weaknesses:**

Strengths:

1.  Through extensive evaluation, the authors report superior performance across various architectures (ResNets, EfficientNet, ViT, Swin). For instance, they show better performance compared to OptFS and SCALE using CNN-based models, while demonstrating relatively limited improvement on vision transformer-based models.

2.  The empirical observations are intuitive, and the figures explaining the Adaptive Scaling process are appealing and helpful.

3.  The novelty lies in integrating design philosophies from existing works such as ASH, SCALE, ODIN, and GradNorm.

Weakness:

1. The paper is hard to follow, and at times the authors use terms without defining them. For instance, the notions of OODness, $\bar{Q'}$ leading to Equation 6, “state-of-the-art,” and claims of generalizability are not clearly explained.

2. The proposed method appears largely incremental in novelty. Most of the design components already exist in the literature: perturbation-based methods (ODIN, GradNorm), scaling frameworks (SCALE, ASH), and selection sets R (BATS, LAPS), among others. While the integration of these philosophies is interesting, it does not justify the strong novelty claims made in the paper.

3. The definition of OODness is overly complex, and the paper lacks sufficient motivation for the design of $p$ in Equation 7. The introduction of correction terms, leading to Figure 5 in the appendix, makes the method difficult to follow. In particular, the correction terms $C$ and $Q$ are confusing. Figures 2 and 5 should be presented together for clarity. Moreover, the overall performance improvement from $Q$ to $Q'$ appears incremental.

4. In the introduction, the second contribution claims the use of a single hyper-parameter; however, the method actually involves multiple hyper-parameters: $k_1, k_2, C, \lambda, o, p_{min}, p_{max}, \epsilon$. The authors do not clearly describe how these parameters are selected. Tuning them using ResNet-50 ImageNet benchmark settings constitutes a significant issue in OOD detection, as it implicitly assumes access to OOD datasets for hyperparameter optimization. The lack of transparency regarding hyperparameter selection is concerning.

5.  Although the reported results are strong across evaluation settings, the evaluation protocol appears to be systematically over-tuned for each setup. In Section 5, the authors claim that “near-optimal performance can be achieved by tuning only $p_{max}$, yet the reported results rely on extensively fine-tuned hyper-parameters. Given such over-tuning, repeated claims of state-of-the-art performance are potentially misleading. Other existing methods could likely achieve comparable or better results under similar levels of tuning, raising concerns about generalizability.

6. In Table 10, relative latency is reported, but it is unclear how the numbers cited in the text were derived. For example, the statement “While SCALE and OptFS have similar latency, AdaSCALE’s is 2.91× …” is incomplete and lacks sufficient explanation.

7. The paper lacks any theoretical or statistical analysis.

8. The paper talks about accuracy in Section 5.1, however ASH explicitly designed to maintain the accuracy. The paper claims to maintain the accuracy, but did not experimentally show the accuracy under Adaptive Scaling.

9. In section 4, authors mention ReAct but I am not sure what insights were taken from ReAct and how this insights translate to other models considered for evaluations like Swin-B and ViT.

---

> ### Author Rebuttal · Authors · 2026-03-28
>
> We thank the reviewer for recognizing AdaSCALE's "superior performance," and "intuitive" observations. We correct several misunderstandings first, then address the concerns.
>
> # Misunderstandings
>
> **"Limited improvement on ViTs"** [ SCALE is the *actual predecessor baseline* on which AdaSCALE improves upon. ] Table 3 shows the opposite. AdaSCALE's FPR@95 improvements (near/far) over SCALE on ViT-B-16 are **23.93%/46.10%**, on Swin-B **17.16%/44.79%**. These are architectures where SCALE struggles most, and AdaSCALE provides the **enormous gains**.
>
> **"perturbation-based (ODIN, GradNorm)"** GradNorm never perturbs input.
>
> **"Selection sets $R$ exist in BATS, LAPS"** Our $R$ is a set of *input pixel indices* for image-space perturbation. Neither BATS nor LAPS defines or uses any analogous selection set. There is no connection at all.
>
> **"$Q$ to $Q'$ improvement is incremental"** Table 8: $Q$ achieves 68.90/42.63 FPR@95 (near/far); $Q'$ achieves 61.17/30.11, **gaps of 7.7 and12.5 points**, far from incremental. Sec. 5.2 and Appendix D.1 show that $Q$ alone can cause overconfident scaling that can invert energy ordering ($E_{\text{OOD}} > E_{\text{ID}}$), while $Q'$ prevents this.
>
> **"$C_o$ and $Q$ are both correction terms"** Only $C_o$ is a correction term, $Q$ is the OOD signal.
>
> **"$C_o$ is a hyperparameter"** $C_o$ is a *computed quantity*.
>
> **"ASH maintains accuracy"** The opposite is true: ASH-S@90 drops ImageNet accuracy by 1.15%, ASH-B@65 by 2.01% (Djurisic et al.). SCALE (Xu et al.) documents drops of 1.14%/3.39% on ImageNet/CIFAR-100. If the baseline achieves 76.70%, AdaSCALE retains exactly 76.70%.
>
> # Concerns
>
> **W2 (Incremental novelty)** Static scaling has existed ${\sim}4$ years; this is first work introducing adaptivity. Non-obvious challenges had to be overcome:
>
> - What signal can make scaling adaptive?
> - What signal gives independent OOD info beyond logits and activation patterns (ASH)?
> - Should it be used raw or tamed? Must it combine with another signal to refine it?
> - How to inject this OOD signal into existing scaling? (via percentile p)
> - ASH/SCALE/LTS fail with ViTs/Swin/EfficientNets, what insights fix this? A key open problem in scaling-based OOD detection.
>
> This paper identified a genuine limitation in existing scaling methods overlooked by the community. We note that adaptive scaling may appear straightforward in retrospect precisely because the decomposition yields clean answers but each question was open prior to our work. We kindly ask the reviewer to evaluate contribution based on what was known beforehand, not with the benefit of hindsight.
>
> **W3 (OODness complexity / motivation for $p$)** OODness is simply $Q'$ normalized to $[0,1]$ via eCDF, a single number driving adaptive $p$. From Sec. 4.1: static $p$ limits separability; higher $p$ yields stronger scaling and higher energy; ID samples need higher $p$, OOD lower. Thus $p = p_{\min} + (1 - F_{Q'}(Q')) \cdot (p_{\max} - p_{\min})$ maps low OODness to high $p$. We will revise to clarify $Q$ as primary signal, $C_o$ as sole correction, and simplify terminology.
>
> **W4 (Multiple hyperparameters)** Sec. 5 states: non-percentile hyperparameters ($\lambda{=}10$, $k_1{=}1\%$, $k_2{=}5\%$, $o{=}5\%$, $\varepsilon{=}0.5$) were determined **ONCE** on ResNet-50 (**SEE TABLE 7 for transparency**) and **FIXED across all 22 ARCHITECTURE-DATASET cases**. Only $p_{\max}$ needs to be tuned per setup for **optimal performance**, identical to ASH/SCALE/LTS tuning a single percentile. Table 5 shows tuning $(p_{\min}, p_{\max})$ vs. only $p_{\max}$ yields trivial difference (62.29 vs. 61.17 near-OOD FPR@95). Sensitivity analysis is in paper/appendix suggesting non-percentile hyperparameters could be treated as **constants**. We tune all baselines with de-facto OpenOOD v1.5 protocol.
>
> **W5 (Over-tuned)** Table 5: with only $p_{\max}$ tuned ($p_{\min}$ set to 60) and **rest treated as constants from Table 7**, AdaSCALE outperforms OptFS (near/far)  by ${\sim}$13%/${\sim}$14% FPR@95 and ${\sim}$8%/${\sim}$2% AUROC in ImageNet-1k.
>
> **W6 (Latency)** The $2.91\times$/$1.56\times$ figures are empirically measured wall-clock ratios over multiple trials. Overhead comes from one extra pass (gradient-based perturbation) and top-$k$ operations. Table 10 isolates variable vs. fixed percentile scaling latency across dimensions.
>
> **W7 & W9 (Theory / ReAct insights)** Sec. 4.2 states: we extend ReAct's finding that OOD samples induce abnormally high activations with the observation that *positions* of these activations are unstable under minor perturbations. ID inputs lie on the learned manifold, so small perturbations preserve activation patterns; OOD inputs are off-manifold, so perturbations create effectively different unfamiliar inputs, shifting which neurons fire maximally. Theoretical analysis is left for future work. **See response to Reviewer ouro.**
>
> ---
>
> We hope these clarifications resolve the reviewer's concerns and are happy to discuss further.

---

> > ### Author Rebuttal · Reviewer_JuwQ · 2026-04-01
> >
> > I just realized that my response to their rebuttal (dated 03/31/2026) was not visible to authors due to a visibility issues. It appears that my detailed comments are only visible to the PC, SAC and AC. Now, I understand their frustration expressed below, and I sincerely apologize.
> >
> > I encourage the authors to carefully read both my previous and the following detailed responses. I appreciate the authors for addressing minor clerical issues and misunderstandings. However, these were never the primary concerns. The more significant issues, the ones I consider to be fundamental flaws in the paper are as follows:
> >
> > 1. $\textbf{Overstated Novelty Claim}$: The majority of the design components already exist in the literature: perturbation (ODIN, G-ODIN), gradient attribution (GradNorm, GradOrth), scaling frameworks (SCALE, ASH), adaptive scaling (ATS[1]), and selection sets R of activation units (BAT, LAPS). While the integration is interesting, it does not justify the strong novelty claims. The novelty lies in how and where these components are applied, which does not warrant the claims made in the paper or rebuttal.
> >
> > 2. $\textbf{Theoretical analysis}$: Given that Adaptive Scaling builds heavily on prior OOD approaches, stronger theoretical or statistical justification is expected. The paper relies mainly on intuition and empirical results, which is insufficient to rigorously justify design choices. This concern remains unaddressed in the rebuttal, except as future work.
> >
> > 3. $\textbf{Hyper-parameters}$: In Introduction, the author claims: “We demonstrate the SOTA generalization ….. by tuning mere one hyper-parameter for a given setup.” and in the reported results in Section 5, authors mention “We tune $(p_{min}, p_{max})$ for optimal results in each case. However, near-optimal performance can be achieved by tuning only $p_{max}$ while fixing $p_{min}$ to 60 across all 22 cases, see Table 5”. However, the main results (Tables 2 and 3) are obtained following a colossal level of fine-tuning, apparent in Tables 31 and 32. With such contradicting nature, presenting tables 2/3 to claim generalization is grossly misleading.
> >
> > 4. $\textbf{Non-percentile hyper-parameters}$: $k_1​$, $k_2​$, $\lambda$, $o$, $\epsilon$, are still hyper-parameters, albeit they are fixed in all evaluation settings. The author's denial of such hyper-parameters is a grave concern. As we can in Table 7, authors have tried multiple values of these non-percentile parameters and choose the ones that work best for ResNet evaluation settings. It's an added advantage that these parameters work well across models, but they still are the hyper-parameters that are fine-tuned. It is concerning, when authors claims mere one hyper-parameter in the contribution ( in introduction section)
> >
> > 5. $\textbf{Hyper-parameter fine-tuned Datasets}$: The authors state that the non-percentile hyperparameters were determined using ResNet-50 via OpenOOD search, but it remains unclear how $(p_{min}, p_{max})$ were tuned. The paper does not specify which dataset was used. It should clarify whether an auxiliary validation set, OOD subset, or held-out data was used. Using OOD data for tuning would violate the OOD-free assumption and may lead to data leakage, and must be explicitly disclosed.
> >
> > 6. $\textbf{SOTA Claim}$: Throughout the paper, the authors make blanket claims of state of the art generalization, which are misleading. The proposed method performs better primarily on ResNet family ( ResNet-50, ResNet-101, RegNet-Y-16, ResNeXt-50, and DenseNet-201). Instead of evaluating multiple models from the same family, including modern architectures such as ConvNeXt would provide a more meaningful assessment. The method also appears to struggle on architectures such as ViT and Swin B, where it does not achieve the same level of performance. In the rebuttal, the authors compare primarily against SCALE for these models, even though SCALE is already known to perform poorly in such settings, as reflected in Table 3.
> >
> > 7. $\textbf{ReAct Insights}$: The rebuttal response provided by the authors on this point is a generic statement that is already well established in the manifold learning literature. My concern is specific: the assumption of abnormally high activations for OOD (ReAct) may not hold for ViT/Swin. Empirically, I did not observe such behavior in penultimate layer representations, suggesting limited generalization of this assumption.
> >
> > 8.  $\textbf{Accuracy Claim}$: The accuracy claim should be backed by explicit results. Since the method modifies logits or activations, it is important to show clearly that classification accuracy is preserved, rather than asserting this informally ( this minor issue).
> >
> > Given that the majority of critical issues remain unresolved, I recommend rejection and encourage the authors to improve transparency, generalizability, and positioning for future submission.
> >
> > [1] ATS: Adaptive Temperature Scaling for Enhancing Out-of-Distribution Detection Methods

---

> > > ### Author Response · Authors · 2026-04-01
> > >
> > > # JuwQ is editing reviews SUBSTANTIALLY after our final response.
> > >
> > > Edit history https://openreview.net/revisions?id=lZl7HjB37H
> > >
> > > **ATTENTION (to all readers)**: We have confidentially reported all alarming evidences of **LLM HALLUCINATIONS** and *irrefutably proven* **REVIEW PLAGIARISM** in original and also in the latest version to AC and PC. (Review requires Policy A). We’ll make them public for readers to judge for themselves after confidentiality no longer bounds us.
> > >
> > > *Reviewer's initial response*
> > >
> > > >> Majority of critical flows [typo] are still present in the rebuttals. I would strongly recommend rejecting the paper and encourage the authors to further polish the work for a future submission in terms of transparency, generizability [typo], better presentation and positioning. Among hundreds of typed words, there are bound to be some typos. Special thanks to the authors for catching them.
> > >
> > > *Tone is **alarming**, and **adversarial**, 2 typos in 2 sentences !! - - no chance to clarify*. Tone shifted & review edited, why?
> > > Context: After we boldfaced “multiple factual errors” hinting LLM hallucination (April 2), the reviewer made significant edits/additions in latest review (April 3). See edit histories and timing.
> > >
> > > 1. (a) Reviewer is blindly categorizing ATS as adaptive scaling because “adaptive” and “scaling” words are in the title of that paper. Just like SCALE/ASH, the scaling procedure of ATS is static.  (b) R is a set of input pixel indices in AdaSCALE and R refers to the reject region in BATS, LAPS. There is no connection as BATS, LAPS fall in one line of work (ReAct, OptFS) while AdaSCALE falls in another line of work (SCALE, ASH).
> > >
> > > **We believe stating incorrect things diminishes credibility of the reviewer in judging novelty.**
> > >
> > > Again, gradient attribution is not even needed (See Table 25 and remark in page 4) and perturbation is a very basic component. If we follow that line of reasoning, it would suggest that future work could rarely be considered novel, since most fundamental components have already been used in prior research.
> > >
> > > ---
> > >
> > > 2. AdaSCALE operates in same empirical tradition as its predecessors, none of which provide formal theoretical guarantees, yet all are widely accepted contributions in this space. We believe establishing strong empirical generalization across 22 settings is a meaningful and self-contained contribution at this stage.
> > >
> > > ---
> > >
> > > 3. **Consistent with literature**, we (just like all OOD detection researchers) tune percentile of ALL STATIC SCALING BASELINES from [60, 65, 70, …, 95, 99] range. **Comparison is COMPLETELY FAIR**. Our design has two percentiles. Is it unacceptable to tune two percentiles instead of one? Ok given, let's fix p_\text{min} to lower limit of that range and only tune p_\text{max} (See Table 5) . See the **ACTUAL DIFFERENCE** in results rather than shallow observation of table placement. For ex. (in near-ood), is difference of ~1/2 FPR point really *grossly misleading* (that can refute generalization claim with single hyperparameter) **when predecessor & prior sota methods are lagging behind by more than 15 & 10**?
> > >
> > > ---
> > >
> > > 4. We hope it makes sense now. If not, we are happy to remove that. This is a fixable issue.
> > >
> > > ---
> > >
> > > 5. We follow **most widely adopted** framework (OpenOOD v1 468 citations, v2 268 citations) in rich literature of OOD detection. OpenOOD uses OpenImage-val split. (OpenOOD **is not** search). Vanilla methods like MSP, MLS, etc are only methods that don't require any hyperparameter.
> > >
> > > ---
> > >
> > > 6. SOTA claim: Reviewer *omits* EfficientNet family [+26%/30% over second best (near/far) FPR@95] and downplays differences in design choices among mentioned architectures. Regardless, we still present requested experiments for ConvNextLarge: Near/Far FPR@95 for [SCALE|OptFS|AdaSCALE] = [91/90 | 74 / 44 | **67 / 40**]  **AdaSCALE is still superior**.
> > >
> > > “SCALE/ASH known to perform poorly in such settings” We believe a famous method (ASH/SCALE) “known to perform poorly” would by default qualify as an important open problem. **SCALE is our baseline, where we substantially improve from in ALL cases!**
> > >
> > > “struggle on ViT and Swin B.”  **See Table 3**. AdaSCALE outperforms OptFS (prior SoTA in generalization) in near-OOD in both Vit and Swin by 5 and 3 FPR metric. Also, it remains competitive in far-OOD category with OptFS in both Vit (within ~1 FPR point) and Swin (within ~2 FPR point). We don’t understand what reviewer means by “struggle” in relative context.
> > >
> > > In spite of all this, if the reviewer still prefers, we are happy to remove the claim of “SOTA generalization”.
> > >
> > > ---
> > >
> > > 7. ReAct is inspiration, not what is used directly. See response to Reviewer auro.
> > >
> > > ---
> > >
> > > **Without even giving us chance to respond, reviewer overconfidently recommended strong rejection.**
> > >
> > > # In light of countless factually wrong statements, LLM hallucinations, and review plagiarism, we sincerely call upon the attention of AC to this matter.

---

### Official Review · Reviewer_dWrw · 2026-03-13

**Soundness:** 4
**Presentation:** 3
**Significance:** 3
**Originality:** 3
**Overall Recommendation:** 5
**Confidence:** 3

**Summary:**

This paper proposes an adaptive scaling method that adjusts the percentile threshold for OOD detection based on estimated OOD likelihood. The OOD likelihood is inferred from the activation shift of high-magnitude activations under minor perturbations—OOD samples exhibit more pronounced shifts than ID samples—and a correction term to mitigate overconfident estimations. Experiments are conducted across 10 architectures and 3 datasets (ImageNet-1k, CIFAR-10/100) demonstrate its state-of-the-art performance.

**Compliance With Llm Reviewing Policy:**

Affirmed.

**Key Questions For Authors:**

* Since the proposed method already has an OODness metric, can we directly use the OODness for detection? Why do you choose to adjust the scaling factor rather than directly detect OOD samples? Can you reduce the latency via direct OOD detection? I look forward to further analyses and empirical results.

* Is the sample-wise adjustment necessary? I wonder whether the scaling factor can be tuned either across the whole dataset or class-wise to achieve similar performance.

**Limitations:**

Yes

**Strengths And Weaknesses:**

# Strength

* The identification that OOD samples exhibit more significant activation shifts under minor perturbations than ID samples is a valuable empirical observation. It provides novel insight.

* Extensive experiments are conducted. The authors conduct comprehensive experiments across a wide range of architectures (e.g., ResNet-50, ViT-B-16, EfficientNetV2-L) and datasets (ImageNet-1k, CIFAR-10/100), demonstrating consistent performance gains in near-OOD, far-OOD, and full-spectrum OOD (FSOOD) detection.

* The paper provides a thorough review of related works and clearly differentiates AdaSCALE from static scaling methods and other post-hoc approaches.

# Weakness

* While the paper has provided detailed results on latency and computational overhead, it seems that the proposed method significantly increases the computational overhead with an additional forward pass and the calculation of OODness.

Generally, I think this paper is well-written with solid analyses and empirical results.

---

> ### Author Rebuttal · Authors · 2026-03-29
>
> We thank the reviewer for the thoughtful and constructive questions. We address each concern below.
>
> ---
>
> ## Q1: Can we directly use the OODness metric ($Q'$) for detection instead of adjusting the scaling factor? Can this reduce latency?
>
> **$Q'$ alone is not sufficiently discriminative; its power lies in being an independent, complementary signal injected through the scaling mechanism.**
>
> - **$Q$ without scaling is not effective.** As we discuss in Section 5, when $Q$ is directly used for near-OOD scoring on ImageNet-1k with ResNet-50, it achieves FPR@95$\downarrow$/AUROC$\uparrow$ of only **79.81/72.32**. In contrast, incorporating $Q$ into the adaptive scaling mechanism substantially improves performance to **59.43/78.14**, a gain of over 20 points in FPR@95. This clearly shows that $Q$ alone cannot serve as a standalone detector; its value lies in providing an independent and complementary OOD signal that is distinct from activation-pattern and logit-based information.
>
> - **The design rationale is rooted in signal fusion.** Scaling-based methods succeed because they *fuse* activation-pattern information with logit-based information (Section 4.1, Pros analysis). $Q'$ provides a *third* independent OOD signal. Using it in isolation discards the rich discriminative information already present in the logits and activation patterns. By injecting $Q'$ through the adaptive percentile (Eq. 7), we achieve a principled three-way fusion where each signal compensates when others fall short.
>
> - **Regarding latency:** with random pixel perturbation (which performs comparably to gradient-based perturbation, see Section F.5), the extra pass for computing gradients is eliminated entirely, reducing AdaSCALE's total latency overhead to **1.56$\times$** over SCALE. This is a modest cost for the significant performance gains.
>
> ---
>
> ## Q2: Is sample-wise adjustment necessary? Could dataset-level or class-wise tuning of the scaling factor achieve similar performance?
>
> **Sample-wise adaptation is critical; dataset-level approaches cannot capture the per-sample variation that drives AdaSCALE's gains.**
>
> - **Dataset-level scaling is exactly what existing baselines do, and it is the limitation we address.** Methods like SCALE and LTS use a *fixed* $p^{th}$ percentile threshold across all test samples (Section 4.1, Cons analysis). AdaSCALE's improvement over these baselines directly quantifies the benefit of moving from a static to a sample-adaptive threshold.
>
> - **Class-wise tuning is infeasible at test time.** OOD detection is performed on individual test samples whose class membership is unknown. Determining whether a sample belongs to *any* known class is the very task we are solving. Class-wise scaling would require either (a) first classifying the sample and then scaling, which introduces circular dependency, or (b) access to labeled OOD data for tuning, which violates the OOD detection setup.
>
> - **Robustness to ID statistics further validates the design.** Table 9 shows that as few as **10 ID validation samples** suffice to construct an effective eCDF, confirming that AdaSCALE's sample-wise mechanism is both principled and practical. It does not require extensive per-class or per-dataset calibration.
>
> ---
>
> We hope these clarifications and empirical references address the reviewer's concerns. We are happy to run any additional experiments if needed.

---

> > ### Author Rebuttal · Reviewer_dWrw · 2026-04-03
> >
> > The rebuttal has addressed my concerns.

---

### Decision · Program_Chairs · 2026-04-30

**Decision:**

Accept (regular)

**Comment:**

This paper proposes an adaptive scaling method for ood detection. The overall idea is intuitive and empirical improvements have been demonstrated. However, this is largely an empirically effective methods. There are concerns over the lack of theoretical analysis, comparisons with more state-of-the art works. The authors are recommended to incorporate the additional results into the paper for the later version to further improve the quality of this paper.